# MnO_2_-Ir Nanowires: Combining Ultrasmall Nanoparticle Sizes, O-Vacancies, and Low Noble-Metal Loading with Improved Activities towards the Oxygen Reduction Reaction

**DOI:** 10.3390/nano12173039

**Published:** 2022-09-01

**Authors:** Scarllett L. S. de Lima, Fellipe S. Pereira, Roberto B. de Lima, Isabel C. de Freitas, Julio Spadotto, Brian J. Connolly, Jade Barreto, Fernando Stavale, Hector A. Vitorino, Humberto V. Fajardo, Auro A. Tanaka, Marco A. S. Garcia, Anderson G. M. da Silva

**Affiliations:** 1Departamento de Engenharia Química e de Materiais-DEQM, Pontifícia Universidade Católica do Rio de Janeiro (PUC-Rio), Rua Marquês de São Vicente, 225 Gávea, Rio de Janeiro 22453-900, RJ, Brazil; 2Departamento de Química, Centro de Ciências Exatas e Tecnologias, Universidade Federal do Maranhão (UFMA), Av. dos Portugueses, 1966 Vila Bacanga, São Luís 65080-805, MA, Brazil; 3Departamento de Química Fundamental, Instituto de Química, Universidade de São Paulo, Av. Prof. Lineu Prestes, 748, São Paulo 05508-000, SP, Brazil; 4Department of Materials, Henry Royce Institute, University of Manchester, Manchester M13 9PL, UK; 5Centro Brasileiro de Pesquisas Físicas, Rio de Janeiro 22290-180, RJ, Brazil; 6South American Center for Education and Research in Public Health, Universidad Norbert Wiener, Lima 15108, Peru; 7Departamento de Química, Instituto de Ciências Exatas e Biológicas, Universidade Federal de Ouro Preto, Campus Morro do Cruzeiro, Ouro Preto 35400-000, MG, Brazil

**Keywords:** manganese dioxide, nanowires, iridium, ORR, low metal loading

## Abstract

Although clean energy generation utilizing the Oxygen Reduction Reaction (ORR) can be considered a promising strategy, this approach remains challenging by the dependence on high loadings of noble metals, mainly Platinum (Pt). Therefore, efforts have been directed to develop new and efficient electrocatalysts that could decrease the Pt content (e.g., by nanotechnology tools or alloying) or replace them completely in these systems. The present investigation shows that high catalytic activity can be reached towards the ORR by employing 1.8 ± 0.7 nm Ir nanoparticles (NPs) deposited onto MnO_2_ nanowires surface under low Ir loadings (1.2 wt.%). Interestingly, we observed that the MnO_2_-Ir nanohybrid presented high catalytic activity for the ORR close to commercial Pt/C (20.0 wt.% of Pt), indicating that it could obtain efficient performance using a simple synthetic procedure. The MnO_2_-Ir electrocatalyst also showed improved stability relative to commercial Pt/C, in which only a slight activity loss was observed after 50 reaction cycles. Considering our findings, the superior performance delivered by the MnO_2_-Ir nanohybrid may be related to (i) the significant concentration of reduced Mn^3+^ species, leading to increased concentration of oxygen vacancies at its surface; (ii) the presence of strong metal-support interactions (SMSI), in which the electronic effect between MnO_x_ and Ir may enhance the ORR process; and (iii) the unique structure comprised by Ir ultrasmall sizes at the nanowire surface that enable the exposure of high energy surface/facets, high surface-to-volume ratios, and their uniform dispersion.

## 1. Introduction

The increasing energy demand, reinforcement of actions for the global warming struggle, rising energy prices due to population growth, and daily power consumption are worrisome [1,2]. Moreover, the depletion of fossil fuel reserves to meet energy requirements is in the spotlight due to energy security worries and political issues [3]. Thus, much effort is currently being dedicated globally to developing cleaner and renewable energy technologies [4,5]. In this scenario, fuel cells, in which ORR is the reaction occurring at the cathode, are considered clean and renewable electrical energy generators with potential for future applications [3,6].

In general, ORR mainly occurs through two pathways, one involving the transfer of two electrons (2e^−^ ORR), producing H_2_O_2_, and the other consisting of the transfer of four electrons (4e^−^ ORR), yielding H_2_O as the product [7,8]. The latter is desirable for practical applications since the 4e^−^ ORR rate is faster and can provide higher energy conversion efficiency [6,8]. Therefore, developing electrocatalysts to undergo this pathway is critical for the optimal electrochemical performance of fuel cells. However, the ORR limits the performance of many electrochemical devices due to its sluggish kinetics, which results in considerable overpotentials, causing a significant loss in energy efficiency; thus, the design of transition metal-based nanomaterials is highly widespread. Although Pt-based electrocatalysts’ low abundance and high cost prevent large-scale implementation on emergent energy conversion devices, they are the most effective cathodes for ORR [9,10]. Therefore, it is essential to develop alternative electrocatalysts with high catalytic activity, availability, and low cost.

Manganese dioxide (MnO_2_) is a promising candidate as an electrocatalyst for ORR because, in addition to the low cost associated with the abundance in the form of natural ores, it has environmental compatibility, low toxicity, and variable oxidation states [11]. However, its major limitation is the low electronic conductivity, which is unfavorable for rapid electron transfer during the electrochemical process [12]. To overcome this limitation, researchers have focused on manipulating the MnO_2_ physicochemical features by controlling its size, shape, composition, and structure (α-, β-, γ-MnO_2_). In this case, it has been demonstrated that controlling the nanoparticle size and shape (e.g., nanocubes, nanorods, nanowires) strongly affects their properties, making nanoparticle shape-control an efficient strategy for maximizing performance [13]. Furthermore, one-dimensional (1D) MnO_2_ nanowires are especially interesting for ORR due to their (i) higher specific surface areas relative to those of commercial supports, (ii) crystal growth along highly catalytically active crystallographic directions, and (iii) surface, which is easily accessible by gas and liquid substrates due to porosity [14].

A strategy to solve the MnO_2_ low electronic conductivity is to deposit small amounts (<2.0 wt.%) of some transition metals. Among them, iridium (Ir) can be considered a promising candidate due to its higher availability than Pt, lower costs, and suitable activities for the ORR [15,16]. However, although some examples have demonstrated that pure Ir-based catalysts or MnO_2_ nanomaterials may be auspicious alternatives for ORR, separately [12,17], they still have several limitations, and deeper investigations by coupling MnO_2_ and Ir are missing in recent literature.

This paper describes the catalytic activity of MnO_2_ nanowires decorated with ultrasmall Ir nanoparticles (NPs) (NPs having diameters of 2–3 nm or less) towards the ORR. Owing to their unique 1D morphology comprised of ultrasmall NPs, we found that the MnO_2_–Ir NPs displayed comparable catalytic performances under low Ir loadings (1.2 wt.%) relative to commercial 20.0 wt.% Pt/C. Interestingly, the MnO_2_–Ir NPs were synthesized by a facile approach based on utilizing MnO_2_ nanowires as physical templates for Ir deposition without any prior surface modification/functionalization steps, as several immobilization processes over oxides require [18,19]. This method enabled the uniform deposition of monodisperse NPs over the entire surface of the support (MnO_2_).

## 2. Experimental Section

### 2.1. Materials and Instrumentation

The analytical-grade chemicals manganese sulfate monohydrate (MnSO_4_·H_2_O, 99%, Sigma-Aldrich Co., St. Louis, MO, USA), potassium permanganate (KMnO_4_, 99%, Sigma-Aldrich Co., St. Louis, MO, USA), polyvinylpyrrolidone (PVP, Sigma-Aldrich Co., St. Louis, MO, USA, M.W. 55,000 g/mol), ethylene glycol (EG, 99.8%, Sigma-Aldrich Co., St. Louis, MO, USA), sodium borohydride (NaBH_4_, 98%, Sigma-Aldrich Co., St. Louis, MO, USA), Pt/C (Pt on Vulcan XC-72, E-TEK), Vulcan XC-72 (E-TEK, Milan, Italy), and Iridium(III) chloride hydrate (IrCl_3_·xH_2_O, 99.8%, Sigma-Aldrich Co., St. Louis, MO, USA) were used as received. All solutions were prepared using deionized water (18.2 MΩ cm). Scanning electron microscopy (SEM) images were obtained using an LEO 440 SEM operated at 5 kV. Four transmission electron microscopes (TEM) were used during this investigation, including: a HITACHI HT 7800 TEM (Tokyo, Japan) operated at 120 kV and a Tecnai FEI G20 TEM, a Jeol ARM 200F TEM, and FEI TALOS F200A with an X-FEG and SuperX (4 SDDs), operated at 200 kV. The nanomaterials for TEM analysis (Tecnai FEI G20, Hillsboro, OR, USA and HITACHI HT 7800 microscopes, Tokyo, Japan) were prepared by a drop-casting approach, in which the NPs suspension diluted in water was deposited over a carbon-coated Tecnai FEI copper grid, followed by drying under ambient conditions. The nanomaterials for high-resolution (HR)TEM and scanning transmission electron microscopy (STEM), and X-ray energy dispersive spectroscopy (XEDS) analysis (Jeol ARM, microscope, and FEI TALOS F200A, Hillsboro, OR, USA) were prepared by making a suspension containing the NPs in isopropanol, followed by their ultrasonic agitation and a drop spread on a TEM Cu grid covered with an amorphous holey lacey film. In the present study, STEM-XED spectrum image datasets were acquired with a dwell time of 100 μs per pixel and a total acquisition time of 900 s. All Talos STEM-XEDS data were processed using Velox 2.3 software (Thermo Fisher Scientific, Waltham, MA, USA) with theoretical k-factors.

Textural characteristics for the electrocatalysts were determined from nitrogen adsorption isotherms, recorded at a relative pressure range of 0.07 < P/Po < 0.3. The Barrett-Joyner–Halenda (BJH) method determined the average pore diameter from the N_2_ desorption isotherms. BET measurements were made in triplicate. The samples’ crystalline structure was analyzed by X-ray diffraction (XRD) using a Rigaku-Miniflex II diffractometer in a 2θ range from 10 to 75° performed at 1°. Before the H_2_-TPR measurements, the samples were heat-treated at 200 °C for 30 min under N_2_ flow to remove moisture adsorbed. After cooling to room temperature, the reduction gas of 8.0% H_2_/N_2_ at a flow rate of 20 mL/min was introduced, and the temperature was raised linearly to 950 °C at a heating rate of 10 [20,21,22,23,24,25,26,27,28,29,30,31] Using Spectro Arcos equipment, the Ir atomic percentages were measured by inductively coupled plasma optical emission spectrometry (ICP-OES). For ICP-OES, the MnO_2_ powder samples were fixed on stainless steel holders using double adhesive carbon tape. X-ray photoelectron spectroscopy (XPS) measurements were performed using an ultra-high vacuum (UHV) chamber with a base pressure of 5 × 10^−10^ mbar equipped with a SPECS analyzer PHOIBOS 150. XPS spectra were acquired using a monochromatic Al-Kα X-ray radiation source (hν = 1486.6 eV). The chemical analyses of the corresponding peak components were performed using CASA software using Shirley-type background and Mn 2p and 3s, and Ir 4f regions line shapes of standard Gaussian–Lorentzian (GL) and Functional Lorentzian (LF), respectively. Direct analysis of the O 1s region was not explored due to the presence of adventitious carbon and Cl, N, and K species related to the precursors on the sample surface. The survey and high-resolution spectra were obtained using energy passes (E_pass_) of 50 and 20 eV, respectively. The spectrometer was previously calibrated using a clean Ag (111) single-crystal with Ag 3d_5/2_ peak position at 368.4 eV and energy resolution given by 0.6 eV. We have not observed any significant charging effect for the MnO_2_ samples. Moreover, each obtained spectrum was calibrated, referencing the adventitious carbon binding energy given by 284.8 eV.

### 2.2. Synthesis of MnO_2_ Nanowires

A hydrothermal approach obtained the MnO_2_ nanowires [20]. In a typical procedure, 0.4 g of MnSO_4_·H_2_O and 1.0 g of KMnO_4_ were dissolved in 30 mL of deionized water. This solution was transferred to a 100 mL Teflon-lined stainless steel autoclave. The autoclave was heated and stirred at 140 °C for 19 h and then allowed to cool down to room temperature [14]. The nanowires were washed three times with ethanol (15 mL) and three times with water (15 mL) by successive rounds of centrifugation and removal of the supernatant and finally dried at 80 °C for 6 h in air.

### 2.3. Synthesis of MnO_2_ Nanowires Decorated with Ir NPs (MnO_2_–Ir NPs)

Typically, 40 mg of MnO_2_ nanowires and 13 mg of polyvinylpyrrolidone (PVP) were added to 10 mL of EG. The obtained suspension was transferred to a 25 mL round-bottom flask and kept under vigorous stirring at 90 °C for 20 min. Then, 1 mL of a 120 mM NaBH_4(aq)_ and 2 mL of a 24 mM IrCl_3_^−^_(aq)_ solutions were sequentially added to the reaction flask. This mixture was kept under vigorous stirring for another 1 h to produce MnO_2_–Ir NPs, washed three times with ethanol (15 mL) and water (15 mL) by successive rounds of centrifugation at 6000 rpm for 5 min and removal of the supernatant. After washing, the MnO_2_–Ir NPs were suspended in 40 mL of water. The same procedure was used to prepare the Ir/C (Vulcan XC-72) electrocatalyst.

### 2.4. Electrochemical Studies

Electrochemical experiments for ORR were performed in a conventional three-electrode cell using a PGSTAT 302 N (Autolab) potentiostat/galvanostat model controlled by Nova 2.0 software and the electrode rotation rate by a Pine ASR rotator. A 0.1 mol L^−1^ potassium hydroxide (KOH) solution was used as the electrolyte, modified glassy carbon as the working electrode, a platinum wire as the counter electrode, and saturated AgCl/KCl as the reference electrode. A total of 20 μL of paint (mixed solution containing 5 mg of the material of interest, 1 mg of methanol with 0.1 mL of Nafion^®^ 5.0% by weight, and 1.4 mL of deionized water, dispersed by ultrasound for 10 min) was used to modify the surface of the glassy carbon electrode. The curves for the oxygen reduction reaction were recorded using the rotating disk electrode (RDE) with different rotation rates, and the electrolyte was saturated with O_2_. All tests were performed at room temperature. The catalytic activity for the ORR in alkaline solution was evaluated by cyclic voltammetry (CV), linear scanning voltammetry (LSV), and RDE techniques. The prepared catalyst was compared to Pt/C commercial electrocatalysts.

The characteristic LSVs obtained using the RDE were described by the Koutecky–Levich (K-L) equation. Considering a first-order kinetics according to the dissolved oxygen, the K-L equation describes the disc current of an RDE system. In that case, the currents are related to the following equation:$$\frac{1}{i}=\frac{1}{i_{k}}+\frac{1}{i_{d}}$$
where *i_k_* is the kinetic current and *i_d_* is the diffusion limiting current, given by:$$i_{d}=0.20nFAD^{2/3}C_{0}\nu^{-1/6}\omega^{1/2}=nB\omega^{1/2}$$
where *F* is 96,485 C mol^−1^, *n* is the number of electrons in the overall reaction, *A* is the disc area (~0.2 cm^2^), *D* is the diffusion coefficient of O_2_ (1.76 × 10^−5^ cm^2^ s^−1^), *C*_0_ is the solubility of O_2_ (1.103 × 10^−6^ mol L^−1^), *υ* is the kinematic viscosity (1.01 × 10^−2^ cm^2^ s^−1^), and *ω* is the rotation rate in revolutions per minute (rpm). Thus, a straight-line plot of 1/I vs. 1/√*ω* for various potentials provides intercepts corresponding to *i_k_*_,_ and the slopes yield the *B* values.

The electrochemically active surface area (EAS) was performed as follows: the sample was saturated with carbon monoxide by bubbling the gas for 5 min at −0.50 V vs. Ag/AgCl (sat). After that, the solution (0.1 mol L^−1^ KOH) was purged for 10 min using N_2_ to remove CO from the solution. The CO monolayer was oxidized by applying two scans at 25 mVs^−1^ in the potential range of −0.5–0.8 V vs. Ag/AgCl (sat).

## 3. Results and Discussion

Our studies started by synthesizing MnO_2_ nanowires through a hydrothermal approach. The procedure afforded well-defined nanostructures of 34 ± 5 nm in width and >1 µm in length (Figure 1A). In sequence, the MnO_2_ nanowires were directly employed as physical templates for the nucleation and growth of Ir NPs on their surface without needing any surface modification/functionalization steps. We used IrCl_3_^−^_(aq)_ as a metallic precursor, PVP as a stabilizer, BH_4_^−^_(aq)_ as a reducing agent, and EG as a solvent.

Figure 1B–F shows SEM (Figure 1B), TEM (Figure 1C,D), and HRTEM (Figure 1E,F) images of the ultrasmall Ir NPs deposited on MnO_2_ nanowires. A uniform distribution of monodisperse ultrasmall NPs with a narrow size distribution all over the nanowires’ surface can be observed, as shown by the histogram displayed in Appendix A (particles’ size of 1.8 ± 0.7 nm). The HRTEM images of individual ultrasmall Ir NP at the MnO_2_ surface (Appendix A) show that the Ir NPs were single-crystalline bounded by {111} facets of Ir fcc structure. High-angle annular dark field (HAADF)-STEM images and STEM-XEDS analyses were performed to investigate further the structure and Mn, O, and Ir elemental distributions in the MnO_2_ decorated with Ir NPs, as shown in Figure 2. Bright-field STEM (Figure 2A–C) and HAADF-STEM (Figure 2D–F) images illustrate the uniform distribution of ultrasmall Ir NPs at the MnO_2_ nanowires support. No significant agglomeration was detected, which often leads to detrimental catalytic properties. In addition, STEM-EDS elemental mapping (Figure 2G–I) confirmed the uniform deposition of ultrasmall Ir NPs over the outer surface of the MnO_2_ nanowires. No morphological changes in the nanowire shape could be detected after Ir NPs deposition.

Then, we focused on investigating how the textural, morphological, electronic, and catalytic performances may vary after reducing ultrasmall Ir NPs at the nanowires’ surface. The textural properties of MnO_2_ and MnO_2_ decorated with Ir NPs obtained by Brunauer–Emmett–Teller (BET) analysis are shown in Appendix A. The specific surface area values for the MnO_2_-Ir nanowires were slightly higher than that of the MnO_2_ nanowires (130 ± 4 vs. 123 ± 6 m^2^/g) due to the intrinsic high surface area of ultrasmall Ir NPs. ICP-OES analyses performed for the MnO_2_-Ir electrocatalyst indicated 1.2 wt.% of Ir loading. The X-ray diffractograms of MnO_2_ and MnO_2_-Ir nanowires (Figure 3A) presented typical reflections assigned to the (100), (200), (310), (211), (301), (411), and (600) planes of α-MnO_2_ (with following lattice constants: a = 9.7847 Å, c = 2.8630 Å, JCPDS 44-0141). No peaks assigned to fcc Ir could be detected, agreeing with their ultrasmall sizes and low loadings in the sample.

Due to metal-support interactions, the deposition of noble metals over MnO_2_ has been an efficient strategy for improving electrocatalytic performances [21,22]. Moreover, metal NPs may interact/cooperate with metal oxides to facilitate catalytic reactions, in which the reducibility of the inorganic matrix usually represents a key element over the detected catalytic activities [23,24]. To further provide in-depth information into these features, the ultrasmall Ir NPs decorated onto MnO_2_ nanowires by hydrogen temperature-programmed reduction was investigated, as shown in Figure 3B. Typically, the MnO_2_ reduction process to MnO can be divided into two reduction stages with Mn_2_O_3_ and Mn_3_O_4_ as intermediates [25,26]. Interestingly, the TPR profile for MnO_2_ nanowires indicates an intense and main reduction peak, centered at 337 °C, which can be assigned to the reduction of Mn^4+^ to Mn^3+^. Furthermore, a shoulder around 340 °C was observed and is related to the reduction of Mn^3+^ into Mn^2+^. These two almost overlapped reduction peaks suggest the coexistence of Mn_2_O_3_ and Mn_3_O_4_ intermediates, corresponding to the combined reduction of MnO_2_/Mn_2_O_3_ into Mn_3_O_4_ and Mn_3_O_4_ to MnO [27,28,29]. In this context, the deposition of ultrasmall Ir NPs at the MnO_2_ surface has tremendously modified its reducibility property compared to the profile for pure MnO_2_ nanowires. More specifically, the reduction peaks for MnO_2_-Ir nanowires were drastically shifted to lower temperatures, exhibiting a primary low intense and broad peak centered at 105 °C, attributed to the reduction of MnO_2_ to Mn_2_O_3_, and a main and intense reduction peak, centered at 165 °C, assigned to the reduction of Mn_2_O_3_ to MnO. This shift to lower temperatures indicates a facilitated reduction process, often related to strong metal-support interactions between metal NPs and MnO_2_ that may occur in the MnO_2_-Ir nanowires [14,29]. Moreover, the position of the hydrogen consumption peaks could also indicate the level of surface oxygen mobility in the metal oxide [30,31]. In this case, the increase in the surface oxygen mobility can also be associated with the shift of the reduction peaks to lower temperatures [30,31], indicating that higher reducibility of the MnO_2_-Ir nanowires relative to pure MnO_2_ nanowires may lead to higher mobility of surface oxygen vacancies species, which can consequently improve its catalytic activity.

XPS analyses were performed to further study this behavior, the surface composition, oxidation state, and the charge transfer tendencies between ultrasmall Ir NPs and MnO_2_ nanowires, as shown in Figure 4. Specifically, we were interested in probing the surface chemical changes that possibly could be observed before and after the deposition of Ir NPs at the nanowires’ surface. The survey spectrum shown in Figure 4A indicates that the samples’ surface is characterized by manganese oxide and iridium, as well as adventitious carbon and small amounts of chlorine, nitrogen, and potassium species related to the precursors. In Figure 4B,C, high-resolution spectra of pristine MnO_2_ nanowires are shown, in which the 4.8 eV splitting energy of Mn 3s and the Mn 2p peak envelope shape evidenced the formation of well-defined MnO_2_ [32]. Surface chemical analysis of MnO_2_ decorated with Ir NPs in Figure 4D indicates that Ir 4f photoemission peak can be described by three distinct doublet components with Ir 4f_7/2_ located at 62.5, 61.8, and 61 eV, related to IrCl_x,_ IrO_2_, and Ir species, respectively [33]. Interestingly, although our analysis suggests a large ratio of oxidized species and IrClx precursor-contamination as compared to metallic Ir, we expect, based on electron microscopy measurements shown in Appendix A, that the Ir NPs display metallic Ir rich-cores.

More important, Figure 4E indicates that the Mn 3s’ larger separation energy splitting is related to MnO_2_ nanowires’ surface reduction from Mn^4+^ to Mn^3+^ [34]. Additional evidence of the MnO_2_ support chemical reduction is shown in Figure 4F with the Mn 2p high-resolution spectrum. In this, the Mn 2p_3/2_ region is analyzed based on multiplet fitting previously described by Biesinger & co-workers [32], in which the component located at 640.7 eV is related to low-coordinated Mn^3+^ species. The increase of the Mn^3+^ species from the pristine MnO_2_ surface compared to the Ir-decorated nanowires suggests oxygen vacancies formation may be promoted due to the Ir NPs synthesis. To estimate the XPS surface composition of MnO_2_-Ir nanowires, we employed the peaks of Mn 3s and Ir 4f7/2, as they are well separated by ~20 eV. For the 1.2 wt.% MnO_2_-Ir nanowires determined by ICP-OES, the Ir 4f7/2/Mn 3s ratio corresponded to 0.14 or 14%. The much higher Ir contents in XPS data relative to ICP-OES can be explained by the fact that XPS is a very sensitive surface technique, in which almost Ir atoms are expected to be exposed at the surface of the nanocatalyst and lead to a higher Ir/Mn ratio.

The ESA of the MnO_2_-Ir nanowires was estimated by CO stripping. From the first voltammogram (Appendix A), the charge for CO oxidation was calculated; usually, the second run recovers the original voltammogram in the pure supporting electrolyte, indicating the complete oxidation of the CO monolayer. The catalyst area was estimated as 2.2 cm^2^. We are aware of the errors in this procedure, and, thus, we did not mean to compare this area value with other commercial electrocatalysts available here.

Owing to their morphology comprising ultrasmall Ir NPs, large surface areas (one-dimensional nanowires displaying a high aspect ratio), the presence of oxygen vacancies, and strong metal-support interactions, we decided to investigate the catalytic activities of the MnO_2_-Ir nanowires towards the ORR; before the tests, CV curves were obtained to evaluate the electrochemical behavior of the electrocatalysts. Figure 5 shows the curves (from −0.5 to 0.2 V) for the glassy carbon modified with bare MnO_2_ nanowires and the prepared MnO_2_-Ir electrocatalyst in an N_2_-saturated aqueous solution KOH 0.1 mol L^−1^ (vs. Ag/AgCl (sat)). The modified electrode with the bare MnO_2_ nanowires presented evident redox waves. Specifically, the reduction peak at −0.35 V is attributed to Mn(IV)O_2_ to Mn(II), whereas the oxidation peak at 0.07 V denotes the oxidation of Mn(II) to MnO_2_ [35]. The CV curve for the modified electrode with MnO_2_-Ir electrocatalyst shows that the shape and electrochemical events are similar to the observed for the bare MnO_2_ nanowires, which can be explained by the low Ir loading (according to the ICP-OES analysis), hampering its identification in the voltammetry. However, a gain in current was detected, which can be related to the heterojunction between Ir NPs and MnO_2_ nanowires, in accordance with microscopy results.

To shed some light on this matter, we tested both materials’ catalytic activity towards ORR (Figure 6A,B). LSVs for the ORR for both materials were recorded at different electrode rotation rates (from 400 to 2500 rpm) in an O_2_-saturated 0.1 mol L^−1^ KOH solution. As expected, the faster the rotation rates, the easier the flow of O_2_ to the electrode surface, followed by the increase in current densities due to the shortened diffusion layer. One can notice that both materials exhibit excellent adherence to the glassy carbon electrode surface. Furthermore, when MnO_2_ nanowires and MnO_2_-Ir electrocatalyst were compared with the commercial Pt/C electrocatalyst (Figure 6C), with a similar metal loading at a rotation rate of 1600 rpm, one can notice that the MnO_2_-Ir nanowires presented a lower onset potential and higher limiting current compared to the Pt/C catalyst, showing its incredible efficiency. We also compared the MnO_2_-Ir nanowires with the 20.0 wt.% Pt/C (Appendix A), which presented current and potential density close to the Pt counterpart. The results were presented in current, without considering the mass of precious metal. However, such a commercial electrocatalyst offers 20 times more metal loading than the MnO_2_-Ir electrocatalyst produced in the present work, showing the effectiveness of our material. The polarization curve of the MnO_2_-Ir material is analogous to the Pt/C counterpart, suggesting that the electrocatalyst can catalyze a 4e^−^ ORR process per reagent molecule. The Koutecky–Levich (K–L) graphs for both materials (Figure 6D) at different potentials displayed a linear relationship, suggesting a first-order reaction in a potential range of −0.50 to 0.20 V.

The K-L plot slopes were used to estimate the number of electrons (n) necessary for the reaction; the MnO_2_-Ir nanowires presented 3.74 e^−^, an approximate 4e^−^ process, whereas the MnO_2_ nanowires presented 3.17 e^−^ during the ORR. The MnO_2_-Ir nanowires presented a similar number of electrons to the commercial 20.0 wt.% Pt/C electrocatalyst (3.89 e^−^). The catalyst based on Pt with similar metal loading to our electrocatalyst presented 2.77 e^−^, which is quite interesting. It does mean that our electrocatalyst can generate H_2_O as the product even at low Ir loading, which is not the case for the commercial catalyst based on Pt with a similar loading. Thus, our data show that the nanoengineering of electrocatalysts enables us to get materials with electrocatalytic results very similar to Pt.

Additionally, Tafel plots were used to offer some understanding of the reaction mechanism of the prepared material. Appendix A displays the Tafel plot and suggests that the rate-determining step corresponds to the first electron transfer once the slopes for the MnO_2_-Ir electrocatalyst was 119.0 mV dec^−1^, and 120 mV dec^−1^ was obtained for the Pt/C, very similar to the literature [36]. The following proposed mechanism is based on our understanding:$$\left[Ir.MnO_{2}{]}^{-}+O_{2}↔[O_{2}-[Ir.MnO_{2}\right]{]}^{-}$$
$$[O_{2}-\left[Ir.MnO_{2}\right]{]}^{-}+e^{-}\to [O_{2}^{-}-\left[Ir.MnO_{2}\right]{]}^{2-}\;\left(determining\;reaction\;step\right)$$
$$[O_{2}^{-}-\left[Ir.MnO_{2}\right]{]}^{2-}+e^{-}+H_{2}O\to [HO_{2}^{-}-\left[Ir.MnO_{2}\right]{]}^{2-}+OH^{-}$$
$$[HO_{2}^{-}-\left[Ir.MnO_{2}\right]{]}^{2-}+2e^{-}+H_{2}O\to [Ir.MnO_{2}{]}^{-}+3OH^{-}$$

Fuel cells in which methanol is used as the fuel are promising due to their high energy density and storage (known as Direct Methanol Fuel Cells—DMFCs). In addition, methanol utilization prompts quick refueling and can be considered portable power systems applications [37,38]. However, Pt-based electrocatalysts are prone to the so-called methanol crossover effect, which is the transport of methanol from the anode to the cathode, lowering the limiting current density of the fuel cell [39]. In this scenario, it was decided to evaluate our prepared electrocatalyst’s resistance to methanol poisoning. Figure 7 depicts the results obtained in the electrocatalytic process towards ORR in the presence of methanol, obtained at 1600 rpm, v = 5 mV s^−1^, O_2_-saturated 0.1 M KOH solution). As a comparison, the Pt/C electrocatalyst was used. One can notice that the Pt/C material presented a significant effect due to the methanol available in the medium at 0.05 M, being not selective and allowing both methanol oxidation and oxygen reduction reactions to occur simultaneously (Figure 7A). Conversely, the MnO_2_-Ir electrocatalyst was very resistant to methanol oxidation, being very selective (Figure 7B). The onset potentials for the electrocatalysts were −0.005 and −0.571 V for the Pt/C and MnO_2_-Ir electrocatalysts, respectively, i.e., even counting on 20 times less metal loading, the Ir-based electrocatalyst was very efficient. The Pt electrocatalyst configuration can explain it; literature deals with the need for three adjacent Pt sites for the methanol oxidation reaction to occur [40,41]. Based on our findings, the same does not happen with Ir active sites. The halfwave potential (E_1/2_) obtained between diffusion limiting regions and the kinetic currents of the polarization curves for the electrocatalysts is similar: −0.19 and −0.21 V for the Pt/C and Ir-MnO_2_ electrocatalysts, respectively. Such data suggests similar reaction kinetics for both [42].

To further evaluate the MnO_2_-Ir electrocatalyst’s efficiency and superior performance, we prepared an Ir/C material using the same procedure. Appendix A shows that such material presents a higher onset potential and a lower limiting current than the MnO_2_-Ir and Pt/C electrocatalysts, offering lower performance. Such a result corroborates the strong interactions between the support and Ir NPs, suggested by the TPR results.

## 4. Conclusions

In conclusion, the present investigation demonstrated that MnO_2_ nanowires decorated with Ir NPs could be employed as heterogeneous electrocatalysts for the oxygen reduction reaction, being a promising nanocatalyst compared to commercial platinum. Through a simple synthesis, it was possible to obtain nanowires with defined shape and size that could serve as templates for Ir NPs nucleation and growth without any surface modification, which in turn oxidize to IrO_2_ while MnO_2_ is reduced. Moreover, the resistance to the methanol crossover effect is significant and opens up the possibility of future studies aiming to improve such an effect for more concentrated methanol solutions.

## Figures and Tables

**Figure 1 nanomaterials-12-03039-f001:**
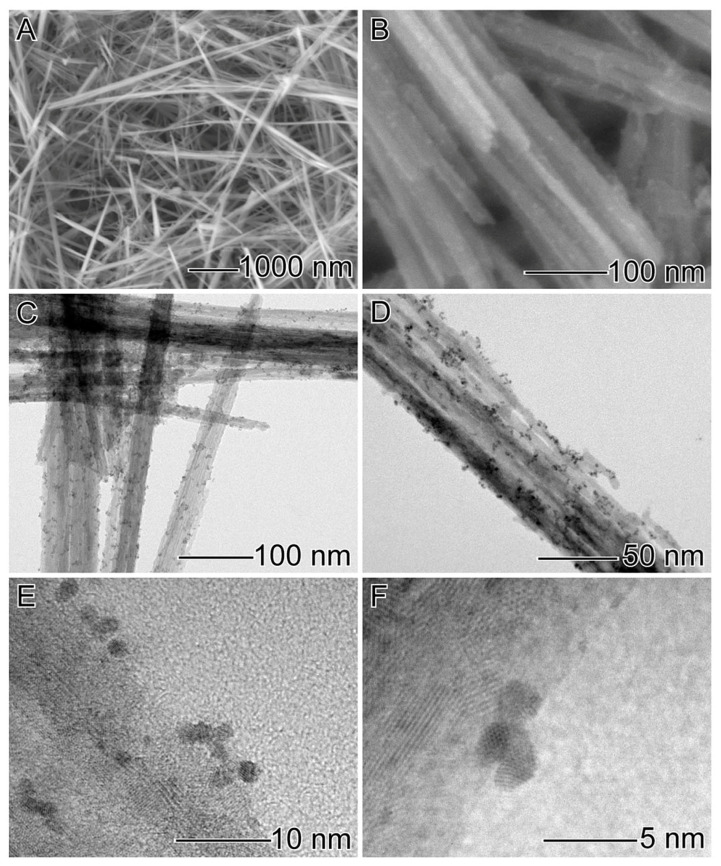
SEM images of MnO_2_ nanowires: (**A**) and MnO_2_-Ir nanowires (**B**); TEM (**C**,**D**); and HRTEM (**E**,**F**) of MnO_2_-Ir electrocatalyst.

**Figure 2 nanomaterials-12-03039-f002:**
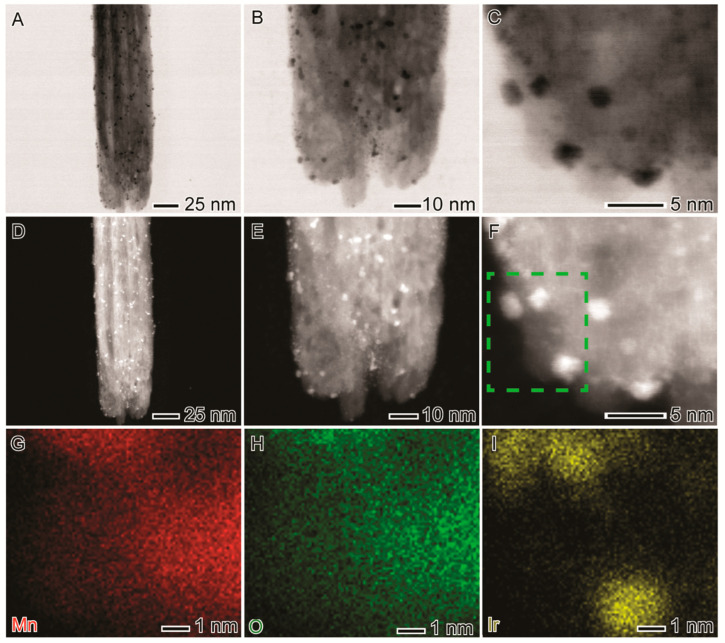
(**A**–**C**) BF-STEM, (**D**–**F**) HAADF-STEM, and (**G**–**I**) STEM-EDS maps of Mn (red, **G**), O (green, **H**), and Ir (yellow, **I**) of the MnO_2_-Ir electrocatalyst.

**Figure 3 nanomaterials-12-03039-f003:**
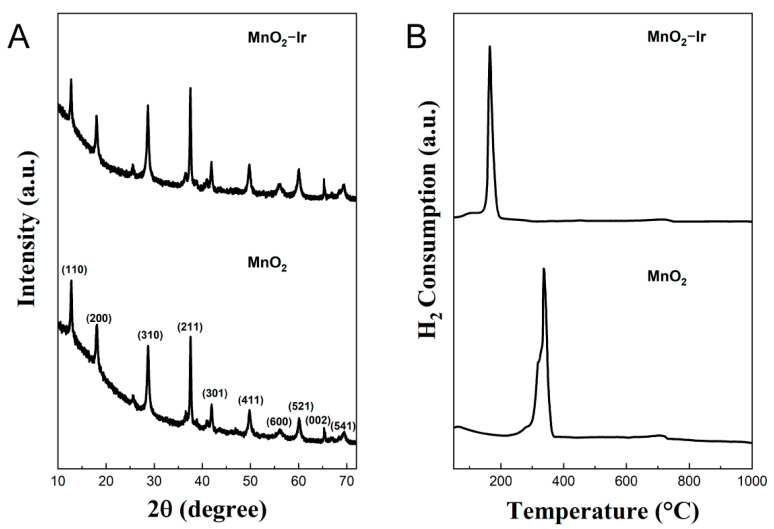
(**A**) XRD patterns and (**B**) TPR profiles of α-MnO_2_ nanowires before and after the deposition of ultrasmall Ir NPs.

**Figure 4 nanomaterials-12-03039-f004:**
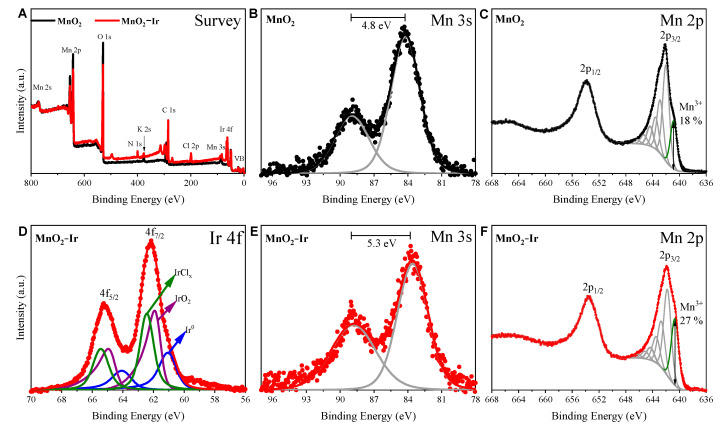
(**A**) XPS survey spectra of MnO_2_ and MnO_2_-Ir (black and red curves, respectively). (**B**,**E**) Mn 3s for MnO_2_ (**B**) and MnO_2_-Ir (**E**), (**C**,**F**) Mn 2p for MnO_2_ (**C**) and MnO_2_-Ir (**F**), and Ir 4f for MnO_2_-Ir (**D**) core-level spectra.

**Figure 5 nanomaterials-12-03039-f005:**
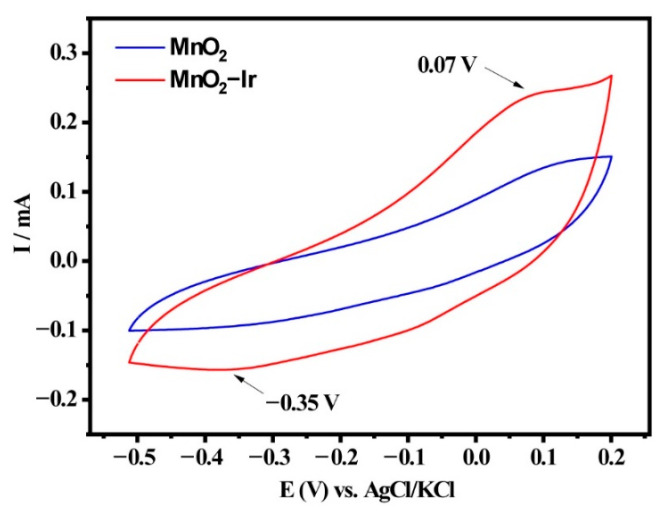
CV curves of the glassy carbon electrode modified with the bare MnO_2_ nanowires and the MnO_2_-Ir electrocatalyst in an O_2_-saturated 0.1 mol L^−1^ KOH solution at room temperature and a scan rate of 25 mV/s.

**Figure 6 nanomaterials-12-03039-f006:**
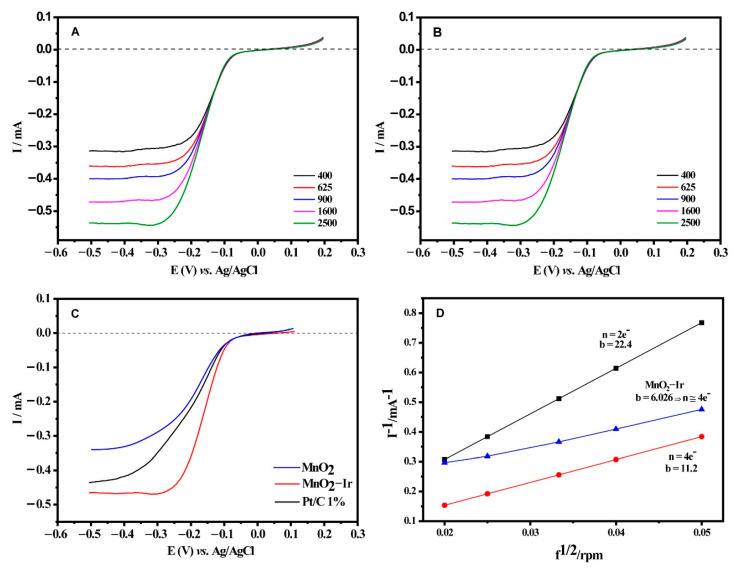
LSVs for the MnO_2_ nanowires (**A**) and MnO_2_-Ir electrocatalyst (**B**) in 0.1 mol L^−1^ KOH solution saturated with O_2_, v = 5 mV s^−1^. (**C**) Polarization curves for the ORR on MnO_2_, MnO_2_-Ir, and 1.2 wt.% Pt/C materials, in 0.1 mol L^−1^ KOH solution, f = 1600 rpm, v = 5 mV s^−1^, room temperature. (**D**) Koutecky - Levich graphs for the oxygen reduction reaction on MnO_2_-Ir and MnO_2_, in 0.1 mol L^−1^ KOH solution, at different electrode rotation speeds, v = 5 mV s^−1^, room temperature.

**Figure 7 nanomaterials-12-03039-f007:**
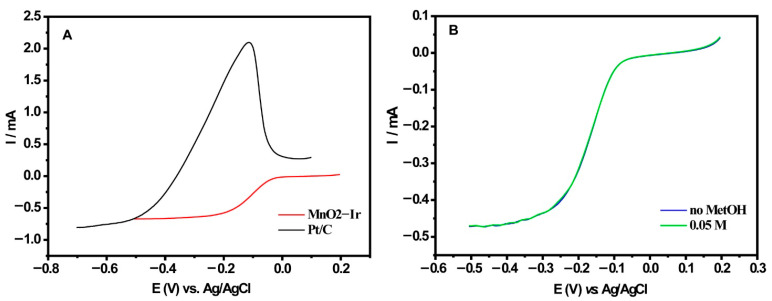
(**A**) Comparison between the ORR polarization curves of MnO_2_-Ir and Pt/C electrocatalysts and (**B**) ORR polarization curves recorded for the MnO_2_-Ir electrocatalyst without methanol and at 0.05 M of the alcohol.

## Data Availability

Data are contained within the article.

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
