# Peer review of "MnO2-Ir Nanowires: Combining Ultrasmall Nanoparticle Sizes, O-Vacancies, and Low Noble-Metal Loading with Improved Activities towards the Oxygen Reduction Reaction"

_nanomaterials, 2022, doi:10.3390/nano12173039_

Round 1

Reviewer 1 Report

The authors reported the synthesis of MnO2-Ir nanowires and demonstrated their improved ORR activities. Their results could be useful to design high-performance catalysts. However, the mechanism for enhanced catalytic performances of MnO2-Ir is still elusive and convincing although the authors proposed four possibilities but without enough experimental or simulation evidence to support or determine the denominating factors. Therefore, I would recommend a major revision before publishing. My specific comments are: 

1. The authors proposed the increased oxygen vacancies in Ir-decorated MnO2 nanowires from the Mn 3s and 2p XPS spectra ( Fig. 4). It is indirect evidence. Are the O XPS spectra different in MnO2 and Ir-MnO2 nanowires?

2. in the abstract, line 44, the authors mention “ the presence of a mix of Ir0 and Ir2+ sites, which are key elements to redox catalytic processes”. The authors should provide some experimental evidence. Besides, the authors should specify the role of Ir0 and Ir2+ in the equation on page 14. 

3.  Both  MnOx and Ir could be ORR catalysts, Particularly, ultra-small Ir nanoparticles. What’s the ORR performance of ultra-small Ir nanoparticles alone? This could further test the hypothesis of strong interactions between MnOx and Ir. 

4. The other minor comments: 

a, the authors should index the lattice spaces in n Fig.1E-1F.

b, is there any signal of Ir in Ir-MnO2 XRD patterns?

Author Response

Dr. Winston Yi

Editor of Nanomaterials

MN: nanomaterials-1844603

Dear Dr. Winston Yi

Thank you for your e-mail dated 2022/07/30. We would like to thank the editor and reviewers for their contributions, which we believe improved the quality of our manuscript. We have addressed all the issues highlighted by the reviewers, as shown below. It is hoped that the revised manuscript is now suitable for publication in Nanomaterials.

Reviewer #1:

  1. "The authors proposed the increased oxygen vacancies in Ir-decorated MnO2 nanowires from the Mn 3s and 2p XPS spectra ( Fig. 4). It is indirect evidence. Are the O XPS spectra different in MnO2 and Ir-MnO2 nanowires?"

We thank the referee for her/his comment. We believe that ex-situ characterization of such material system cannot lead to reliable surface chemical evidence through the oxygen peak analysis. Our reasoning is based on the fact that the O 1s spectra have contributions from residual carbon from the atmosphere, as observed on the spectra below label as C-Ox. Therefore, we have not explored this photoemission peak in our analyses. Nevertheless, we have mentioned our reasoning on not performing the O 1s analysis in the amended text in page 7.

  1. "in the abstract, line 44, the authors mention "the presence of a mix of Ir0 and Ir2+ sites, which are key elements to redox catalytic processes". The authors should provide some experimental evidence. Besides, the authors should specify the role of Ir0 and Ir2+ in the equation on page 14."

Thanks for the observation. With the "the presence of a mix of Ir0 and Ir2+ sites, which are key elements to redox catalytic processes" statement, we meant that the existence of Ir0 and Ir2+, i.e., different redox species, is an important feature for electrochemical processes, with the possibility of reversibility routes. Unfortunately, we cannot prepare completely reduced or completely oxidized Ir-based materials to show the importance of such species due to the necessity of exposing the material to the air for the electrochemical performance study. Thus, we decided to remove the sentence from the manuscript.  

  1.  "Both  MnOx and Ir could be ORR catalysts, Particularly, ultra-small Ir nanoparticles. What's the ORR performance of ultra-small Ir nanoparticles alone? This could further test the hypothesis of strong interactions between MnOx and Ir."

Thanks for the observation. We prepared Ir nanoparticles over carbon to evaluate its performance, and we obtained a  higher onset potential and lower limiting current compared to the Ir-MnO2 and Pt/C catalysts. Such information was included in our revised manuscript as highlighted in blue on pages 17, 18 and Figure S6 on Support Information.

4a. "The other minor comments: a, the authors should index the lattice spaces in n Fig.1E-1F.

To better demonstrate the crystallographic surface facets exposed, we included in our revised manuscript HRTEM images in higher magnifications and resolutions as highlighted in blue in Figure S2, on Support information.

4b." is there any signal of Ir in Ir-MnO2 XRD patterns?"

No diffraction peaks assigned to any iridium crystalline phase were observed, most probably as a result of the formation of very small nanoparticles on the MnO2 nanowire surface as well as a low metal loading (~1.2 wt%). This was clarified in our revised manuscript as highlighted in blue on page 11.

Sincerely yours,

Prof. Anderson G. M. da Silva  

Reviewer 2 Report

The authors have presented a study regarding a new noble metal free catalyst which should be used in fuel cells, to substitute Pt-based catalyst for the ORR in the cathodic compartment.

The work has been quite deeply presented, even if some crucial points should be addressed in order to let the paper be eligible to be published in Nanomaterials Journal, after major revisions.
The main point, which have risen some doubts about presented results, is that there is a lack in the explanation of the Oxygen vacancies presence. Other lacks have been found here and there in the text. 
I will list what I mean, point by point:

1- In line 217 you mentioned the BET results (130 vs. 123 m2/g): since these two values are quite comparable, it would be great if you can add their uncertaint value, in order to check if they really overlap or not.

2- In fig.4 c & f you have reported Mn2p doublets, together with some sort of deconvolution procedures (only for Mn(+3) component). I would suggest to add, at least for the Mn2p3/2 peaks, the entire set of peaks, as reported by Biesinger et al, in the referenced paper you have mentioned, to show the multiplet splitting due to Mn(+4) contribution.
3- You stated that the Ir4f doublet is related mostly to Ir(0), although m
etal Ir4f7/2 should be located at (60.8 +/- 0.2) eV. Moreover, if it is a metal, the fitting procedure should be performed with an asymmetric line shape (lineshape of LA(1.13,1.9,5)), see http://www.xpsfitting.com/search/label/Iridium. It seems, in my opinion,  that you have found more IrClx, due to the position of your peak maximum, instead of metal Ir. Try to performed a deconvolution considering both aspects, by giving the relative amount of the two species found. Also the relative intensity of the two 4f components seems wrong, due to the relative height of the two peaks (4f5/2 seems too small if a ratio of 0.8 is expected).   

4- You are speaking about O vacancies but you are not showing O1s peaks related to the two samples. You have just mentioned it by looking at the Mn signals..any comments on this?

5- In figure 5 please indicate the position of the peaks mentioned in the discussion, with arrows, for the sake of clarity.

6- Could you please calculate the electron transfer number n from the RRDE measurements (for MnO2, MnO2-Ir and Pt/C samples), as reported in N. Garino et al.  2019 2D Mater. 6 045001 , in order to have more precise numbers, instead of the graph shown in fig. 6-D, where we can evaluate only the range in which your curve is located?

7- In fig.6-D please add a legenda for the used colors, which is different from the one reported in fig.6-C.

Author Response

Please find attached the response letter

Reviewer 3 Report

The article "MnO2-Ir nanowires: Combining Ultrasmall Nanoparticle Sizes, O-vacancies and Low Noble-Metal Loading with Improved Activities Towards the Oxygen Reduction Reaction" presents a method for obtaining a composite material MnO2-Ir nanowires, its structure is detail studied, and catalytic activity is evaluated in the ORR of the received material. The article is of undoubted relevance, since it is devoted to the problem of obtaining highly efficient materials for fuel cells, which are an important part of hydrogen energy. However, a number of important remarks require an answer:

The work compares the activity of MnOx-Ir with Ir loadings 1.2 wt. % and commercial Pt/C with Ir loadings 20.0 wt. %, which is incorrect, since both the loading of metals and the nature of the carrier and the nature of the metal are different. Thus, there are many factors and the positive effect of iridium has not been proven. It is necessary to make a comparison with similarly obtained MnOx-Pt with Pt loadings (about 1.2 wt. %), then it will be possible to show exactly the advantage of Ir compared to Pt.

Another important aspect, when comparing the activity of MnOx-Ir with Ir loadings (1.2 wt. %) and commercial Pt/C (20.0 wt. % of Pt), only current values (mA) are given, and not specific current per mass of precious metal in the catalyst (A/g(Pt) or (Ir)) or per specific surface area (ESA) in A/m2 (Pt) or (Ir). Usually, to compare the activity of catalysts, the values of the half-wave potential and the values of specific currents at the selected potential are used. Note that the article sets the task of reducing the platinum content in the catalyst, and, accordingly, reducing the cost of the catalyst, while today the cost of iridium significantly exceeds the cost of platinum. Therefore, the calculation of activity per mass of precious metal is absolutely necessary.

An important characteristic of catalysts is the value of the electrochemically active surface area (ESA), it is clear that in the case of MnOx-Ir with Ir loadings (1.2 wt. %), such measurements are difficult due to the low loading of the metal, however, the CO-stripping method for measuring the surface area should be tested.

From the XPS data, it is necessary to evaluate the surface composition and compare it with the ICP-OES results.

Based on TEM (or SEM) data, it is necessary to construct a histogram of the nanoparticles size distribution.

Author Response

Dear reviewer,

Please find attached the response letter.

Best regards

Round 2

Reviewer 1 Report

The authors have made the necessary revisions and the article can be accepted for publication.

Reviewer 2 Report

Dear authors,

I have deeply appreciated all the efforts spent in order to fulfill referees requests.

I guess that know the level of your manuscript has been really elevated, and results are now clear and well explained. Now it can be considered suitable to be published as it is in Nanomaterials Journal.

Thanks for your professionalism and work well done.

Reviewer 3 Report

The authors have made the necessary edits and the article can be accepted for publication.